# Effects of Fe-Ions Irradiation on the Microstructure and Mechanical Properties of FeCrAl-1.5wt.% ZrC Alloys

**DOI:** 10.3390/nano11123423

**Published:** 2021-12-17

**Authors:** Runzhong Wang, Hui Wang, Xiaohui Zhu, Xue Liang, Yuanfei Li, Yunxia Gao, Xuguang An, Wenqing Liu

**Affiliations:** 1Institute of Materials, School of Materials Science and Engineering, Shanghai University, Shanghai 200444, China; wrzh1995shu@163.com (R.W.); ZXH21@shu.edu.cn (X.Z.); 2Science and Technology on Reactor Fuel and Materials Laboratory, Nuclear Power Institute of China, Chengdu 610041, China; 3Key Laboratory for Microstructures, Shanghai University, Shanghai 200444, China; liangxue@shu.edu.cn; 4Suzhou Nuclear Power Research Institute, Suzhou 215004, China; p193258@cgnpc.com.cn; 5Key Laboratory of Materials Physics, Institute of Solid State Physics, Chinese Academy of Sciences, Hefei 230031, China; yxgao@issp.ac.cn; 6School of Mechanical Engineering, Chengdu University, Chengdu 610106, China; anxuguang@cdu.edu.cn

**Keywords:** FeCrAl, irradiation damage, ZrC, nanoindentation, atom probe tomography

## Abstract

Fe-13Cr-3.5Al-2.0Mo-1.5wt.% ZrC alloy was irradiated by 400 keV Fe^+^ at 400 °C at different doses ranging from 6.35 × 10^14^ to 1.27 × 10^16^ ions/cm^2^ with a corresponding damage of 1.0–20.0 dpa, respectively, to investigate the effects of different radiation doses on the hardness and microstructure of the reinforced FeCrAl alloys in detail by nanoindentation, transmission electron microscopy (TEM), and atom probe tomography (APT). The results show that the hardness at 1.0 dpa increases from 5.68 to 6.81 GPa, which is 19.9% higher than a non-irradiated specimen. With an increase in dose from 1.0 to 20.0 dpa, the hardness increases from 6.81 to 8.01 GPa, which is an increase of only 17.6%, indicating that the hardness has reached saturation. TEM and APT results show that high-density nano-precipitates and low-density dislocation loops forme in the 1.0 dpa region, compared to the non-irradiated region. Compared with 1.0 dpa region, the density and size of nano-precipitates in the 20.0 dpa region have no significant change, while the density of dislocation loops increases. Irradiation results in a decrease of molybdenum and carbon in the strengthening precipitates (Zr, Mo) (C, N), and the proportionate decrease of molybdenum and carbon is more obvious with the increase in damage.

## 1. Introduction

Cladding materials, as the first barrier for a reactor, have to withstand the high temperature, high pressure, and strong irradiation and corrosion of high-temperature steam. The traditional zircaloy cladding can react violently with steam under high pressure and temperature, and release large amounts of hydrogen and heat, causing a severe hydrogen explosion, which is the main reason for the Fukushima nuclear accident [1,2,3,4,5,6]. There is an urgent need to develop other cladding materials with better performance. FeCrAl alloy, which has no risk of causing a hydrogen explosion, is considered to be a promising candidate for advanced reactor accident-resistant cladding because of its excellent resistance to corrosion and oxidation in high-temperature steam [7].

The researchers have improved the properties of FeCrAl alloy by optimizing the composition, determining its range to be within Fe-(10~14) Cr-(3~5) Al [8,9,10,11,12]. Nevertheless, the disadvantage posed by the poor radiation resistance of FeCrAl alloy cannot be solved by optimizing its composition [13,14,15]. It has been found that introducing nano-sized oxide, carbide, or nitride particles could effectively refine the grains and optimize the grain boundaries of the alloy [16,17]. Moreover, these dispersed nano-precipitates can be sinks for absorbing the interstitials and vacancies induced by irradiation [18]. All of these factors increase the irradiation resistance for FeCrAl alloys as cladding.

Past research about nano-precipitate-dispersion-strengthened FeCrAl alloys has mainly focused on FeCrAl-Y_2_O_3_. However, due to the high aluminum content in FeCrAl alloy and the strong affinity between yttrium and aluminum, a large number of large-size Y-Al-O is generated, which deteriorates the properties of the material. [19,20,21]. ZrC possesses high chemical stability, shows a high melting point (~3540 °C), and does not react with Al, which can effectively avoid the formation of coarse particles. In addition, ZrC obtains a much lower thermal neutron absorption cross-section than Y_2_O_3_ [22]. Therefore, compared with Y_2_O_3_, ZrC is more ideal for reinforcement of FeCrAl alloys. Wang et al. prepared ZrC-reinforced FeCrAl alloys and found that the grains are significantly refined and higher temperature strengths are significantly enhanced compared to the pristine FeCrAl alloy [23]. Wan et al. also found that high tensile strength, large elongation at room temperatures, and high temperatures can be obtained for the FeCrAl alloy with 0.6 wt.% ZrC [24].

The irradiation performance of the FeCrAl-1.0wt.% ZrC alloy has been studied [22]. It is worth noting that Fe-ions irradiation induces the formation of a large amount of fine secondary phase, such as MoC particles and α′ phase. These particles will significantly improve the hardness of the material. Based on the above background, the irradiation properties of the ZrC-reinforced FeCrAl alloys, depending on the dose of Fe-ions irradiation, are further investigated by nanoindentation, TEM, and APT techniques in this paper. The effects of different doses of Fe-ions irradiation on the mechanical properties and microstructures of the reinforced FeCrAl alloys are discussed in detail.

## 2. Experimental Procedures

Fe-13-Cr-3.5Al-2.0Mo-1.5 wt.% ZrC alloy was selected as the irradiation experiment specimen in the present work. Fe-ions irradiation was conducted with 400 keV Fe ions up to a dose of 6.35 × 10^14^ Fe^+^/cm^2^ (peak irradiation dose is 1.0 dpa) and 1.27 × 10^16^ Fe^+^/cm^2^ (peak irradiation dose is 20.0 dpa) at 400 °C, respectively. The irradiation dose rate is 1.0 × 10^12^ Fe^+^/cm^2^·s and the temperature of the specimen was controlled at 400 °C during the whole process.

The nanoindentation tests were conducted using Nano Indenter G200 (KLA Inc., Milpitas, CA, USA). The average hardness was obtained from 10 indentations with a 5 × 2 array and two points with large error are excluded for each specimen. The interval between the two test points was 30 μm to avoid any overlap with the deformation region below each indentation and the limit of indentation depth was 300 nm. Other parameters are given as follows: surface approach velocity was 10 nm/s, harmonic displacement target was 2 nm, frequency target was 45 Hz, poisons ratio was 0.3, and temperature was 20 °C.

A double-beam focused ion beam method (FIB, Helios 600 Nanolab, FEI HONGKONG Co., Ltd., CZ) was utilized to prepare the specimens for transmission electron microscopy (TEM) and atom probe tomography (APT) measurements. The details of TEM and APT specimen preparation using FIB are the same as those in Refs. [25,26].

The microstructure of the irradiated specimen was observed with a JEOL 2100F field emission gun (FEG) TEM (JEOL LTD, Tokyo, Japan) and images were captured and analyzed by Gatan digital micrograph microscopy suite software (3.20.1314.0, Gatan Inc., Pleasanton, CA, USA).

The APT analyses were performed on a LEAP-4000X HR (CAMEMA, Madison, WI, USA) at a temperature of 50 K in the laser mode with an energy of 60 pJ in a vacuum of ~10^−9^ Pa. Reconstruction and analyses were carried out utilizing IVAS 3.6.8 software (CAMEMA, Madison, WI, USA). Field evaporation and detector efficiency were set to 27 V·mm^−1^ and 0.37, respectively. The strengthening precipitate in this alloy is a nitrogen-combining carbide of niobium plus molybdenum, namely, (Zr, Mo) (C, N). The 1.0 at.% Zr iso-concentration surfaces was obtained to highlight the (Zr, Mo) (C, N). In addition, the precipitates induced by irradiation are MoC and α′ phase, and we used 1.5 at.% Mo and 27 at % Cr iso-concentration surfaces, respectively, to define them.

The identification and characterization of (Zr, Mo) (C, N) precipitates, MoC, and α′ phase, which will be mentioned later, were undertaken by the maximum separation method with maximum separation distance (*d*_max_) and minimum solute atom number (*N*_min_). The *d*_max_, *N*_min,_ and other relevant parameters of different precipitates are shown in Table 1. The determination and calibration of all these parameters listed in Table 1 are given in the Appendix A.

## 3. Results and Discussion

### 3.1. SRIM Simulation

Figure 1 shows the profiles of damage and implantation which were predicted by SRIM (Stopping and Ranges of Ions in Matter) for the specimen after 400 keV Fe-ions irradiation. The maximum damage and implantation are located around a depth of 80 nm and 150 nm from the irradiated surface, respectively. In addition, both decrease to zero at a depth of 360 nm after reaching the corresponding peak. Under the two different irradiation doses, the maximum displacement damage is 1.0 dpa (6.35 × 10^14^ Fe^+^/cm^2^) and 20.0 dpa (1.27 × 10^16^ Fe^+^/cm^2^), respectively.

### 3.2. Nanoindentation

Figure 2 shows the profiles of average hardness with indentation depth, which were obtained from specimens at different irradiation doses by nanoindentation. Due to the existence of indentation size effects (ISE), softer substrate effects (SSE), and surface effects, the average hardness decreases with the increase in depth, as shown in Figure 2 [27].

The hardness is a comprehensive reflection of the hardness of the whole plastic influence zone. The radius of the plastic-affected zone is about 5~7 times the indentation depth. Hence, if the press-in depth is beyond a certain value, the unirradiated region will begin to impact the hardness data; namely, the softer substrate effects (SSE) occur.

A model was proposed by Li [28] to explain the SSE. There is a transition point of SSE that appears at the critical depth (*h_c_*), which depends on the irradiation hardening value. Figure 3 shows a method to determine *h_c_* as proposed by Liu [29]. The ratios of *H_irr_*/*H_unirr_* at the same depth were obtained and the peak ratios were located at a depth of about 50 nm. A purple dashed arrow along the peak is used to represent the transition line. When the press-in depth beyond the depth value corresponds to transition line, the SSE starts to influence the accuracy of the measurement of hardness in the irradiated region. By this method, the critical indentation depth *h_c_* was determined to be about 50 nm, which is about one-sixth of the damaged layer’s thickness.

In the shallow irradiation region, the indentation size effect (ISE) significantly affects the hardness measurement result [30,31,32]. For more accurate hardness, a formula was established by Nix and Gao to expounded the ISE based on the Geometrically Necessary Dislocation (GND), assuming that the indenter is of perfect rigidity and the material hardness is three times the yield strength [31]. The Nix–Gao model predicts the hardness depth profile as follows:(1)$$\frac{H}{H_{0}}=\sqrt{1+\frac{h^{*}}{h}}$$
where *H* is the hardness for a given depth of indentation, *H*_0_ is the hardness at infinite depth, *h** is a characteristic length which depends on the material and the shape of the indenter tip, and *h* is the depth of the press-in. Experimental data were re-plotted as *H*^2^ vs. 1/*h* in order to obtain the Nix and Gao parameter *H*_0_, as shown in Figure 4.

It can be seen from Figure 4 that the profiles of the irradiated specimens at different doses show bilinearity and both shoulders are located at a depth of ~60 nm, which is almost consistent with the depth corresponding to the position of the transition line in Figure 3; namely, *h_c_*. This is because, when the active area of the indenter reaches above the irradiation layer, the non-irradiated softer substrate affects the measurement of hardness. However, the data curve of the non-irradiated specimen shows a slight bilinearity with a shoulder depth of ~70 nm as well, which is because of the significant surface effect in the shallow region (<100 nm) of the specimen. Therefore, assuming that the surface effect has the same effect on the specimens at different doses, the *H*_0_ of the three specimens in the range of 40 nm < *h* < 60 nm is calculated by the method proposed by Kasada [32] so as to minimize the effect of surface effect on the hardening irradiation measurement. The hardness and irradiation hardening increment (Δ*H*) of the three specimens are listed in Table 2, respectively, and the average hardness of specimens with damage is shown in Figure 5.

It can be found in Table 2, compared with the 5.68 GPa hardness of the non-irradiated specimen, the hardness of the 1.0 dpa specimen increases to 6.81 GPa, which is about 19.9% higher than the non-irradiated specimen. The irradiation dose increases 20 times, from 1.0 to 20.0 dpa, and the hardness increases from 6.81 to 8.01 GPa, which is only an increase of 17.6%. The increased rate of hardness in the 1.0~20.0 dpa progress is much lower than that in the 0~1.0 dpa progress, indicating that, with the increase in damage, the irradiation hardening of the specimen tends toward saturation. This trend of saturation can be found more intuitively in Figure 5. It must be noted that Figure 5 does not express that there is a clear linear relationship between hardness and dpa, but illustrates that the increased rate of hardness decreases significantly with the increase in dpa. For example, comparing the slopes of two independent straight lines can obviously find this phenomenon.

### 3.3. TEM

TEM was performed to observe the microstructure of the damage region of the high-dose irradiated specimen (1.27 × 10^16^ Fe^+^/cm^2^) and combined with the SRIM simulation results (Figure 6). It is shown that the contrast of the specimen rises with the increase in depth, the reason for which is the formation of a large number of dislocation loops induced by irradiation. At a depth of 80 nm from the surface, the contrast decreases after reaching the peak and there are almost no defects, such as dislocation loops, in the deeper region. The selected area electron diffraction (SAED) pattern in the internal of the specimen is displayed in the upper left corner of Figure 6. The diffraction pattern is single crystal diffraction spots, indicating that no amorphization occurred in the matrix of this specimen at the 20 dpa dose.

According to the SRIM results, damage of the sample irradiated to a high dose is 20.0 dpa at ~80 nm and 1.0 dpa at ~260 nm from the sample surface. Therefore, the observation of microstructure could be conducted in the low damage region (region 1) and high damage region (region 2) in a single specimen The damage levels at the centers of region 1 and region 2 are about 20 dpa and 1 dpa, respectively.

### 3.4. APT

Figure 7 shows the results of atom distribution maps and iso-concentration surfaces of C, Mo, Zr, and Cr obtained at different irradiation doses (0/1.0/20.0 dpa).

As shown in Figure 7a, C, Zr, Mo, and the molecular ions of ZrN^+^ are segregated, while the homogeneous distribution of Cr is observed. Moreover, the segregations of C, Zr, N, and Mo are located in the same position, indicating the formation of nano-sized (Zr, Mo) (C, N) precipitates. The research shows that Mo can improve performance of the second phases in the material [33]. For example, Mo can be incorporated as MoC into (Nb, Mo) (C, N) in microalloyed steel, reducing the driving force for coarsening of second phases [34]. Compared with Nb (C, N) having no Mo, the (Nb, Mo) (C, N) nano-precipitates possess stronger pinning and strengthening ability. Similarly, (Zr, Mo) (C, N), as the nano-sized precipitates with Mo, can also provide a better strengthening effect for FeCrAl alloys.

Based on the simulation of SRIM of specimens with a high irradiation dose (1.27 × 10^16^ Fe^+^/cm^2^), the two different regions, which are a low damage region (~1 dpa) and a high damage region (~20 dpa), are intercepted and analyzed by APT, respectively (Figure 7b,c). It can be seen from Figure 7b,c that, compared with the APT results of non-irradiated specimens, the enrichment positions of Mo and Zr in the Zr-enriched domains no longer completely coincide, and a huge count of tiny Mo-enriched domains are present in the matrix. In addition to the coincidence of C with the enrichment position of Zr, there are also high-density and nano-sized C-enriched domains in the matrix, which correspond to the position of the Mo-enriched domains. The segregation of N and Zr are unchanged and there is no new segregation in the matrix. Moreover, fine, spherical Cr-enriched clusters with high number density, regarding the α′ phase, are generated due to the degree of rising spinodal decomposition induced by irradiation [14,35,36].

The diameter (*d*) and number density (*N*) of various secondary phases at different irradiation doses were calculated (Table 3). In this experiment, the Zr-enriched domain, Mo-enriched domain, and Cr-enriched domain represent (Zr, Mo) (C, N), MoC, and α′ phase, respectively. When calculating the *d* and *N* of MoC, Mo-enriched domains with the same segregation position as Zr-enriched domains were excluded. However, it must be mentioned that the existence of local magnification affects results in the precipitates’ size, obtained by APT, to appear slightly larger than the actual size [37,38]. It can be seen from Table 2 that, with the increase in irradiation damage, the *d* of (Zr, Mo) (C, N) decreases slightly and the *N* remains unchanged. The *d* of MoC increases and the *N* has no change. The *d* of the α′ phase does not change significantly and the *N* increases.

The formation of MoC and α′ phase is one of the reasons for irradiation hardening of materials. To quantify the contribution of precipitation to material hardening, the dispersed-barrier hardening (DBH) (Equation (2)) model was used to calculate the strength increment Δ*σ_y_* caused by nano-precipitates.
(2)$$\Delta \sigma_{y}=f\left(N,d\right)=\alpha MGb\sqrt{Nd}$$
where *α* is the barrier strength of nano-precipitates taken as 0.048 ± 0.012 (α′ phase) and 0.27 ± 0.06 (MoC) [22,39], *M* is the Taylor factor (~3.06 for bcc Fe-based materials) [40], *b* is the Burgers vector (~0.228 nm), and *G* is the shear modulus (~82.2 GPa) [41]. The terms *d* and *N* represent mean diameter and number density of nano-precipitates, respectively.

The increase in strength associated with precipitates induced by irradiation can be calculated by linear-superposition Equation (3) as follows [39]:(3)$$\Delta \sigma_{y}=f\left(N_{\alpha^{\prime }},d_{\alpha^{\prime }}\right)+f\left(N_{MoC},d_{MoC}\right)$$

The increment of strength (Δ*σ_y_*) can be converted to the increment of hardness (Δ*H_n_*) by using the Rice equation [42] as follows:(4)$$\Delta H_{n}=\Delta \sigma_{y}/0.274$$

Based on the above calculations, and ignoring the hardening of pure Fe caused by irradiation, the experimental and calculative data obtained for contributions from the hardness effects for the formation of MoC and α′ phase under irradiation at different doses are summarized in Figure 8. The calculated value of hardness (Δ*H_cal._*) is 1.10 GPa at 1.0 dpa, which is accordant with the experimental data. Nevertheless, the Δ*H_cal_*_._ is 1.41 GPa at 20.0 dpa, which is lower than the corresponding experimental value. The reason for this phenomenon is the formation of loops with high density induced by high irradiation dose, and all of these loops make significant contributions to hardening. The reason corresponds to the TEM results in Figure 2.

The chemical composition of strengthening precipitate (Zr, Mo) (C, N) at different irradiation doses was measured by APT. Single (Zr, Mo) (C, N) precipitates indicated by the black arrow in Figure 7 were selected for one-dimensional concentration distribution analysis (Figure 9). It must be noted that the composition of all (Zr, Mo) (C, N) precipitates in specimens at different irradiation doses is roughly the same, so a single precipitate selected can represent all of them in the one specimen. It can be found that the enrichment of Mo is the same as that of C, Zr, and N in (Zr, Mo) (C, N), though it is not enriched in the outer layer of the particle, which is different from the results of some other studies [43,44]. This indicates that the presence of Mo in (Zr, Mo) (C, N) does not inhibit the coarsening of the second phase by blocking the diffusion of elements to the second phase through the formation of a Mo-enriched layer. According to the APT results, it is speculated that Mo doped into the precipitates as MoC and reduced the surface energy of the secondary phase/matrix, thus reducing the driving force of the coarsening of the secondary phase, which is similar to the research results of Enloe et al. [34].

In addition, with the increase in damage, the proportion of different elements in (Zr, Mo) (C, N) precipitates changes.

As we can see in Figure 9, the contents of Zr, Mo, C, and N in the precipitates are too low to be practical. This is because the existence of trajectory aberrations caused by local magnification effects results in a large amount of Fe and Cr being introduced into (Zr, Mo) (C, N) [37,38]. Hence, all Fe and Cr in (Zr, Mo) (C, N) were removed, while Zr, Mo, C, and N were reserved during the progress of calculation. The proportion of the four elements in (Zr, Mo) (C, N) was calculated and the peak proportions of each element are listed in Table 3.

It can be seen in Table 4 that, before irradiation, the peak proportions of C, Mo, Zr, and N in (Zr, Mo) (C, N) are 38.7%, 27.1%, 19.8%, and 14.4%, respectively. When the irradiation dose is 1.0 dpa, the peak proportions of C, Mo, Zr, and N in the (Zr, Mo) (C, N) are 33.2%, 24.2%, 26.2%, and 16.4%, respectively, and the proportions of C and Mo in the precipitates decrease significantly. When the irradiation dose is 20.0 dpa, the peak proportions of C, Mo, Zr, and N in the (Zr, Mo) (C, N) are 31.2%, 18.8%, 31.1%, and 18.9%, respectively. The peak proportions of C and Mo in the (Zr, Mo) (C, N) continue to decline.

The ratio of (Zr + Mo)/(C + N) in (Zr, Mo) (C, N) is roughly 1:1. Therefore, (Zr, Mo) (C, N) could be seen as the complex of ZrC, ZrN, and MoC. The absolute value of molar free energy of formation of ZrC (Δ*G*_298_ = −193.3 kJ mol^−1^) and ZrN (Δ*G*_298_ = −336.7 kJ mol^−1^) is much higher than the molar free energy of formation of MoC (Δ*G*_298_= −10.0 kJ·mol^−1^). In addition, the melting points of ZrC (~3532 °C) and ZrN (~2952 °C) are also much higher than that of MoC (~2700 °C), indicating that ZrC and ZrN are more stable than MoC. Therefore, irradiation causes MoC in (Zr, Mo) (C, N) to decompose, resulting in a decrease in the contents of Mo and C. With the increase in irradiation damage, the cascade collision increases, resulting in more MoC decomposition in (Zr, Mo) (C, N). Therefore, the specific gravity of Mo and C further decreases, resulting in the size of (Zr, Mo) (C, N) trending down, which is consistent with the results in Table 4.

The reduction of Mo and C in (Zr, Mo) (C, N) would result in the decline of its anti-coarsening ability. Several studies have described that coarsening will occur under the working environment of high temperature and weaken the pinning effect of precipitates for grain boundaries and dislocations, resulting in deterioration of the high temperature properties and anti-irradiation properties of the material [45,46]. With the increase in irradiation damage, more Mo and C in (Zr, Mo) (C, N) diffuse and segregate at the interface of MoC, resulting in an increase in MoC precipitate size, which is also consistent with the results in Table 4.

## 4. Conclusions

Nanoindentation, TEM, and APT were utilized to study the effects of irradiation dose on the microstructure and mechanical properties of FeCrAl-1.5wt.% ZrC alloys. The conclusions of this work can be summarized as follows:Fe-ions irradiation results in significant hardening of FeCrAl-1.5wt.% ZrC alloys. Compared with the 5.68 GPa hardness of an unirradiated specimen, the hardness of an irradiated specimen with a 1.0 dpa irradiation dose increases to 6.81 GPa, an increase of 19.9%. When the irradiation dose increases 20 times, from 1.0 dpa to 20.0 dpa, and the hardness increases from 6.81 to 8.01 GPa, an increase of only 17.6%. The increasing rate of hardness is much lower than that from unirradiated to an irradiation dose of 1.0 dpa, indicating that, with the increase in damage, the hardening of the specimen tends toward saturation.When the irradiation dose is 1.0 dpa, the hardening of the material is mainly caused by nano-precipitates induced by irradiation, such as α′ phase and MoC precipitates. When the irradiation dose is 20.0 dpa, the dislocation loops and nano-precipitates induced by irradiation also occur and, together, lead to increased material hardness.The Fe-ions irradiation enhances the diffusion of Mo and C in (Zr, Mo) (C, N). With the increase in irradiation, the proportion of Mo and C in (Zr, Mo) (C, N) decreases, the size of (Zr, Mo) (C, N) decreases, and the size of MoC increases.

## Figures and Tables

**Figure 1 nanomaterials-11-03423-f001:**
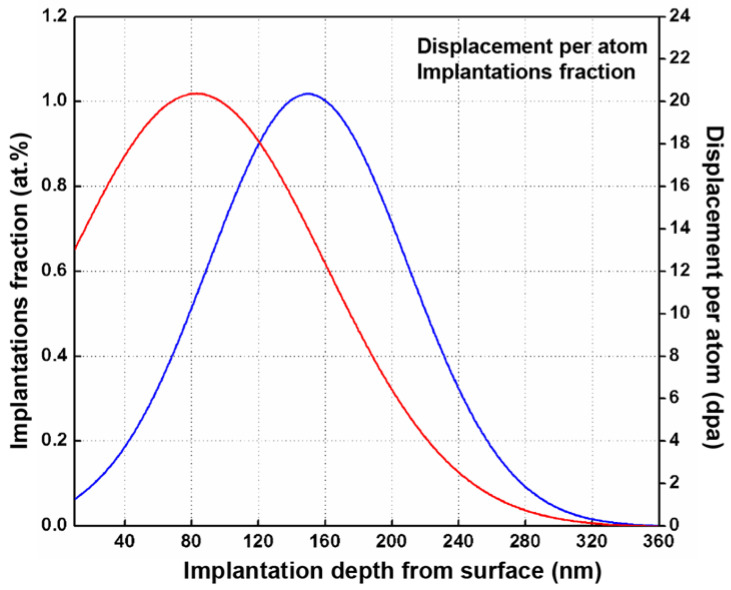
The profiles of damage (dpa) and implantation predicted by SRIM08 code. The red line indicates the change of dpa with indentation depth, and the blue line indicates the change of implantation fraction with indentation depth.

**Figure 2 nanomaterials-11-03423-f002:**
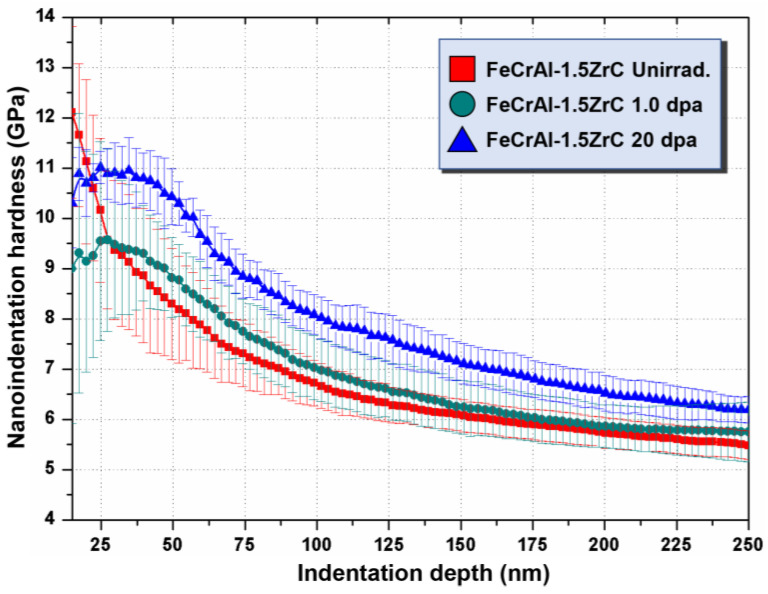
Curves of *H* and *h* for average nanoindentation hardness of specimens at different irradiation doses. The red line, green line and blue line represent the variation of hardness with indentation depth of unirradiated, 1.0 dpa and 20 dpa specimens, respectively.

**Figure 3 nanomaterials-11-03423-f003:**
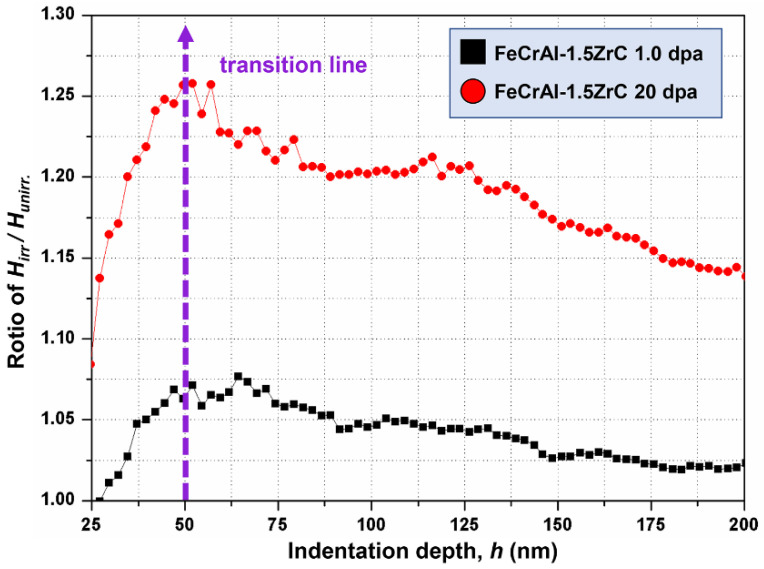
Dependence of the ratio of *H_irr_*/*H_unirr_* on indentation depth at the irradiation doses of 1.0 dpa and 20.0 dpa.

**Figure 4 nanomaterials-11-03423-f004:**
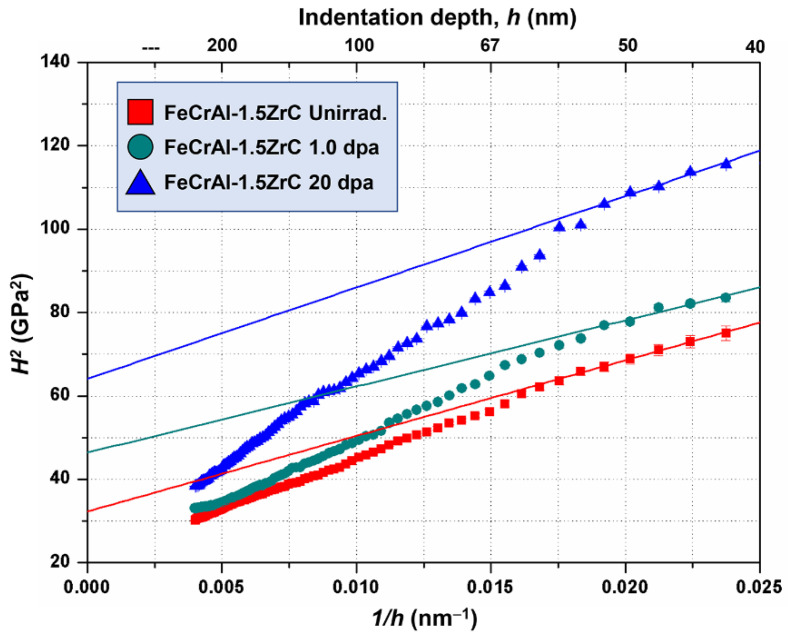
Profiles of *H*^2^ and *h^−^*^1^ for specimens at different irradiation doses. The red, green and blue lines represent unirradiated, 1.0 dpa and 20 dpa specimens, respectively.

**Figure 5 nanomaterials-11-03423-f005:**
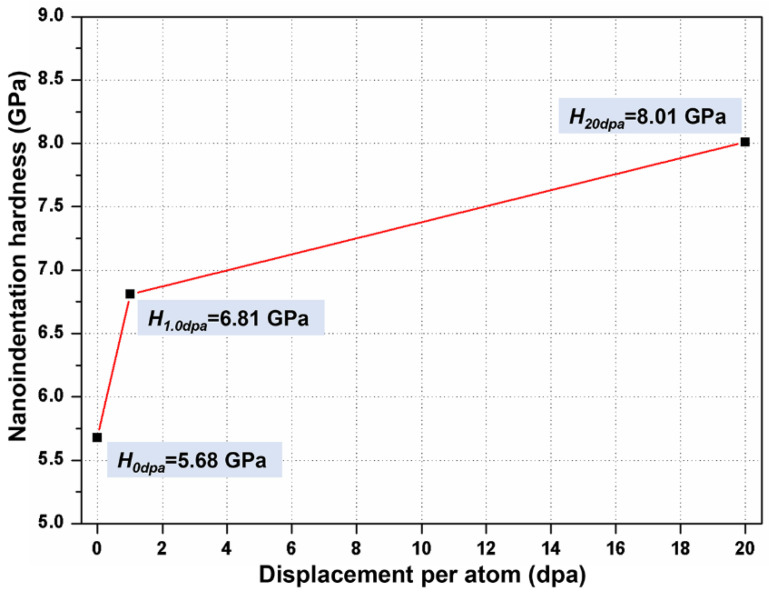
Plots of hardness (*H*) vs. damage for specimens at different irradiation doses.

**Figure 6 nanomaterials-11-03423-f006:**
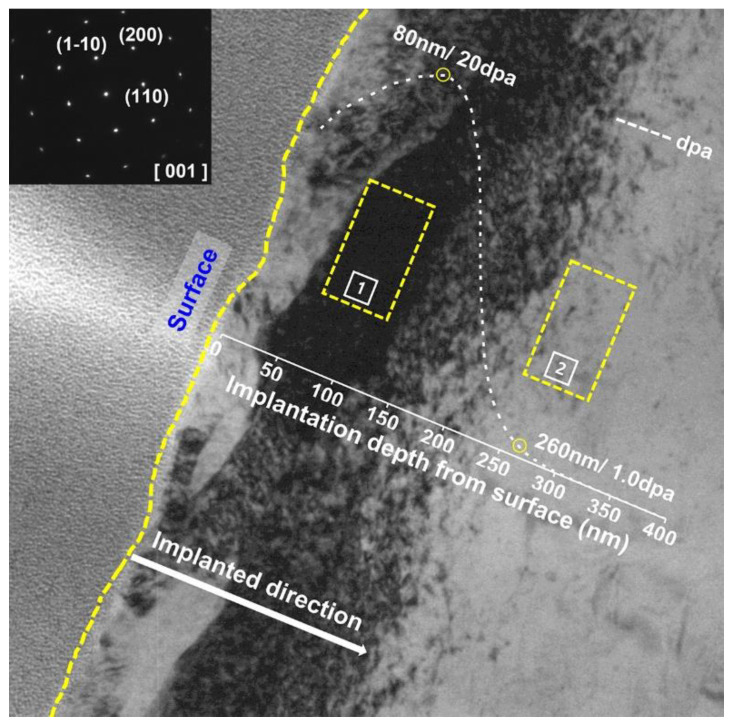
TEM images of the damage regions for the specimen irradiated with 400 keV Fe-ions to 1.27 × 10^16^ Fe cm^−2^ overlaid with the predicted dpa profile by SRIM. Region 1 in the figure is a high damage region and region 2 is a low damage region. The insert figure in the upper left corner is diffraction pattern of matrix.

**Figure 7 nanomaterials-11-03423-f007:**
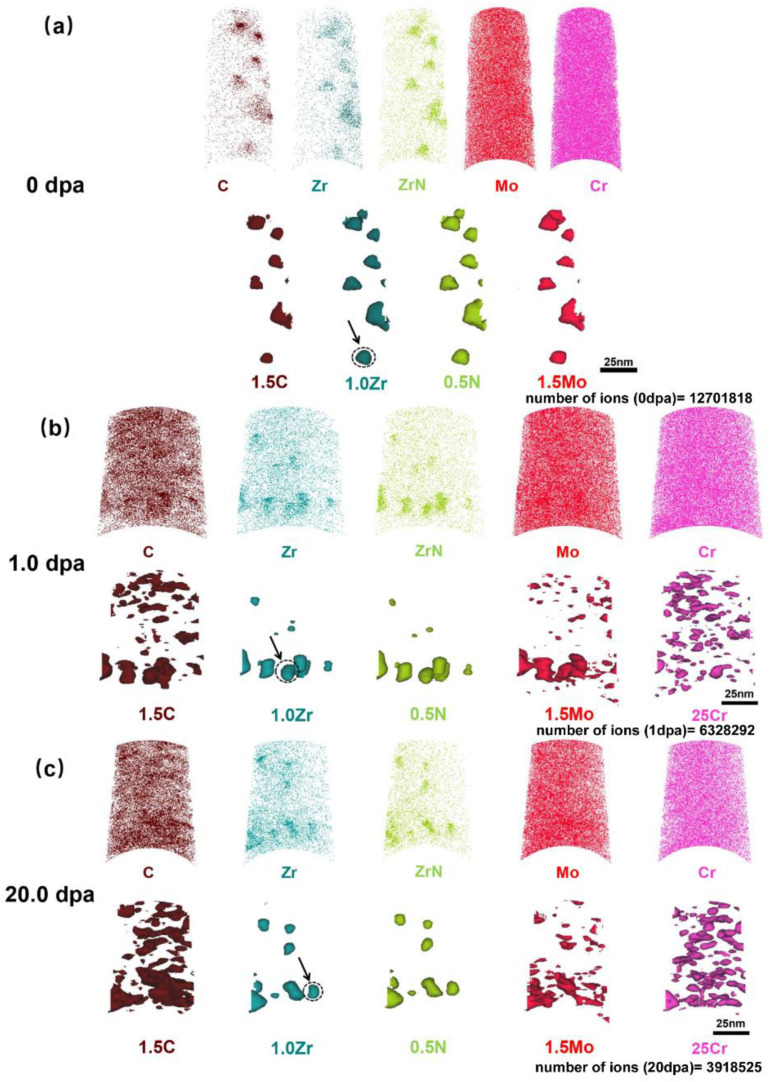
3D distribution maps of C, Zr, ZrN^+^, Mo, and Cr, as well as the iso-concentration surfaces of 1.5 at % C, 1.0 at % Zr, 0.5 at % N, 1.5 at % Mo, and 25 at % Cr, at different irradiation doses measured by APT: (**a**) 0 dpa, (**b**) 1.0 dpa, and (**c**) 20.0 dpa. Since the Cr is distributed homogeneously in the unirradiated specimen, it is not necessary to make iso-concentration surfaces of Cr. The three Zr-enriched domains in the dotted circle, indicated by the black arrow, will be separately selected for analysis later.

**Figure 8 nanomaterials-11-03423-f008:**
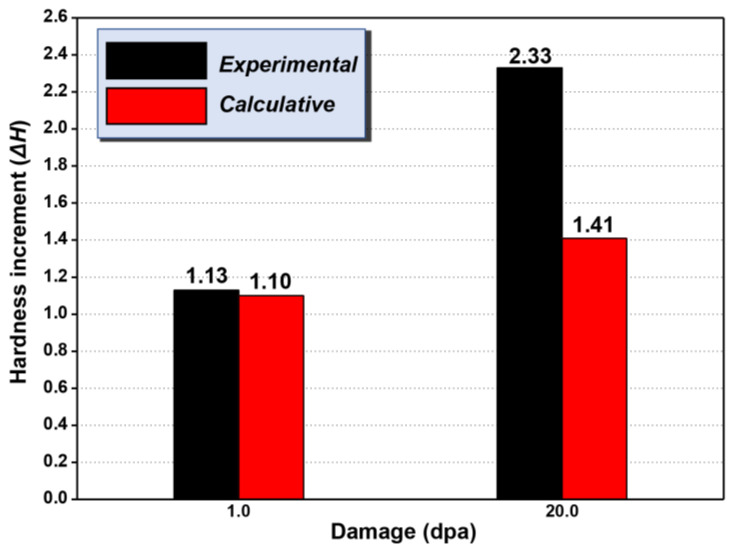
Schematic diagram of the radiation-induced hardness increment (Δ*H*) in the experiment (black) and calculation (red) at the damage dose of 1.0 dpa and 20.0 dpa, respectively.

**Figure 9 nanomaterials-11-03423-f009:**
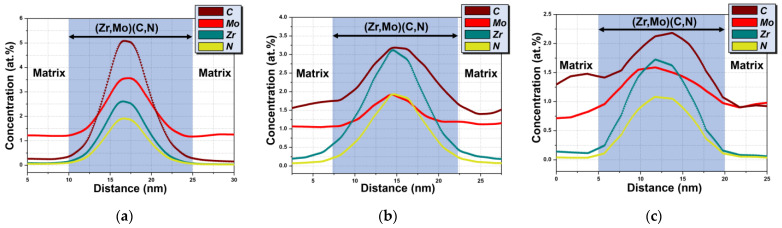
1D concentration distribution of C, Mo, and Zr atoms corresponding to a single (Zr, Mo) (C, N) particle, indicated by the black arrow at different irradiation dose specimens (**a**) 0 dpa, (**b**) 1.0 dpa, and (**c**) 20.0 dpa, respectively.

**Table 1 nanomaterials-11-03423-t001:** The relevant parameters of different precipitates for “Cluster Analysis” in IVAS 3.6.8 software.

Precipitate	*d*_max_/nm	*N*_min_/ions	*Order*/ions	*L*/nm	*d*_erosion_/nm
(Zr, Mo) (C, N)-unirr.	3.2	50	1	3.2	3.2
(Zr, Mo) (C, N)-irr.	2.6	45	1	2.6	2.6
MoC	1.6	40	1	1.6	1.6
α′ phase	0.7	12	1	0.7	0.7

**Table 2 nanomaterials-11-03423-t002:** Calculated *H*_0_ and Δ*H* based on the Nix–Gao model of specimens at different irradiation doses.

Irradiation Dose	*H*_0_/GPa	Δ*H*/GPa
0 dpa	5.68	0
1.0 dpa	6.81	1.13
20 dpa	8.01	2.33

**Table 3 nanomaterials-11-03423-t003:** The diameter (*d*) and number density (*N*) of precipitates in different irradiation doses.

Type of Precipitate	*d*/nm	*N*/ × 10^22^ m^−3^
(Zr, Mo) (C, N) (0 dpa)	5.6 ± 1.1	4.1
(Zr, Mo) (C, N) (1.0 dpa)	5.0 ± 2.0	5.1
(Zr, Mo) (C, N) (20.0 dpa)	4.6 ± 1.2	4.6
MoC (1.0 dpa)	1.8 ± 0.5	30.4
MoC (20.0 dpa)	3.2 ± 0.7	29.7
α′ phase (1.0 dpa)	1.7 ± 0.3	164.3
α′ phase (20.0 dpa)	1.8 ± 0.4	201.1

**Table 4 nanomaterials-11-03423-t004:** Peak proportions of C, Mo, N, and Zr in (Zr, Mo) (C, N) with different irradiation doses.

dpa	C	Mo	Zr	N
	at %			
0	38.7	27.1	19.8	14.4
1	33.2	24.2	26.2	16.4
20	31.2	18.8	31.1	18.9

## Data Availability

Data underlying the results presented in this paper are not publicly available at this time but may be obtained from the authors upon reasonable request.

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
