# Peer review of "Effects of Fe-Ions Irradiation on the Microstructure and Mechanical Properties of FeCrAl-1.5wt.% ZrC Alloys"

_nanomaterials, 2021, doi:10.3390/nano11123423_

Round 1

Reviewer 1 Report

This manuscript needs major revision. The title refers to the Effects of different Fe-ions irradiation, but only two dose values are studied,1 dpa and 20 dpa. The use of 'different dose values', in this case, is misleading.

English grammar and expressions must be revised, specially the repeated use of 'influence' instead of 'fluence'. Some sentences are not easy to understand due to this. 

No experimental details about the nanoindentation tests are given (number of indentations, number of curves measured per sample, maximum load used, etc.); these must be included.

The authors mention that the hardness has reached a saturate at 20 dpa, but this statement seems to be incorrect: 6.81 GPa at 1 dpa represents a 19.9% increase respect to 5.68 GPa in the non-irradiated material; 8.01 GPa at 20 dpa represents a 41.02% increase respect to 5.68 GPa in the non-irradiated material. The mentioned 17.6% increase in hardness compares the value at 20 dpa with the one at 1 dpa, and that does not mean that the hardness value has reached a saturate. In case the authors refer to a saturate in the hardening increase rate, that conclusion cannot be stated by comparing the 0-1dpa and 1-20dpa ranges. 

A large number of dislocation loops is mentioned in the TEM analysis, have the authors made any quantification? Diffraction inset in Fig. 5 is not described in the caption nor in the text. Is there a TEM image from the sample irradiated at 1 dpa?

Details about the number of samples analyzed by APT, number of ions in each dataset, etc. needs to be included. The isosurface values used to define α' precipitates only appears in Fig.6; it should be mentioned in the text. Was any chemical analysis done in the TEM analysis?

The 1D concentration profiles displayed in Fig. 8 correspond to the particle highlighted in Fig. 6. Did the authors analyze other precipitates to confirm this result being representative or to get some statistics for the chemical quantification?

Conclusion 1 needs to be rewritten for the reason previously mentioned. The meaning of the sentence 'Fe-ions irradiation leads to material of the reinforced FeCrAl alloy' is not clear.

Reviewer 2 Report

L71 : Fe13Cr3.5Al2.0Mo-1.5 wt.% ZrC alloy --> Fe-13Cr-3.5Al-2.0Mo-1.5 wt.% ZrC alloy
L73 : A couple of irradiation periods during Fe ion irradiations at 400C should be described.  
L85 : The section on atom probe tomography does not provide any detail about atom probe tomography analysis. The authors need to specify whether it's in laser or voltage mode and the oscillation frequency. In identifying clusters in atom probe tomography, one needs to do a full calibration by testing different Dmin and Nmin values. This is because the use of arbitrary Dmin and Nmin values lead to results with a low degree of confidence. A full calibration needs to be provided in the main body of the paper or appendix to justify the use of the Dmin and Nmin values otherwise the data presented would have a low confidence level. Please consider the following book which is widely recognised by atom probe microscopists.
https://www.springer.com/gp/book/9781461434351 

L154  : The reference [32] should be the following article ;  R. Kasada, Y. Takayama, K. Yabuuchi, A. Kimura, Fusion Eng. & Des. 86 (2011) 2658-2661
L167 : A comment : It is preferred to show the TEM observation result for a deformed area by nano-indentation if possible.
Figure 5 : Data of TEM observation for Fe-ion irradiated specimens with 1dpa should be shown as well as that with 20dpa.  
L181 : How does the author consider the thermal ageing effect for specimens during irradiation? An unirradiated specimen with the same thermal ageing condition as the irradiation experiment condition will be helpful to understand it. I guess 13% Cr content in Fe alloy is not enough to produce the spinodal segregation at 400C but enhanced segregation of alpha' phase could be seen in the results of APT. It is necessary to take account consideration about the thermal ageing effect for the radiation-induced/enhanced segregation.
Figure7 : The fraction of hardening contribution for each defect component should be displayed in the column of calculative data.  The defect components mean the precipitate as ZrMoCN, MoC and alpha' and dislocation loops 
I can not fully understand the microstructural changes in APT measurement after Fe ion irradiation. In the shallower area, ZrMoNC precipitate was dissociated and MoC and a' particle appeared. On the other hand, in the deeper area around a tale of Fe ion rage, ZrMoNC still remained and grew up, and MoC and a' particle also appeared. Is that correct for my understanding? When the deeper area was not formed by irradiation phenomena, MoC and a' particle were formed by thermal ageing effect, were not they?

Round 2

Reviewer 1 Report

I sincerely thank the authors for considering my suggestions and answering to my queries. The manuscript has been very much improved and can now be published. I only have one last comment: In new Figure 5, the experimental points cannot be linked by  straight lines, as the hardness vs. dpa dependance may not be linear and the current display might be suggesting it.

Reviewer 2 Report

I would like to commend the authors for the many efforts they have made to revise the paper.
There is one point that I would like to see corrected. In Table 3, the values of diameter and number density were given in three significant digits, but I wonder if the measurements were that precise. I would like you to correct it as follows.

5.61±1.10 --> 5.6±1.1
4.07 --> 4.1
